

**PeerJ Hubs**
Published on behalf of

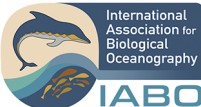

# Opportunistic consumption of marine pelagic, terrestrial, and chemosynthetic organic matter by macrofauna on the Arctic shelf: a stable isotope approach

Valentin Kokarev[1], Anna K. Zalota[1], Andrey Zuev[2], Alexei Tiunov[2], Petr Kuznetsov[3], Olga Konovalova[4,5] and Nadezhda Rimskaya-Korsakova[3]

[1] Laboratory of Ecology of Coastal Bottom Communities, Shirshov Institute of Oceanology RAS, Moscow, Russia
[2] Laboratory of Soil Zoology and General Entomology, A.N. Severtsov Institute of Ecology and Evolution RAS, Moscow, Russia
[3] Department of Invertebrate Zoology, Faculty of Biology, Lomonosov Moscow State University, Moscow, Russia
[4] Centre of Marine Research, Lomonosov Moscow State University, Moscow, Russia
[5] National Research Tomsk State University, Tomsk, Russia

Corresponding author
Nadezhda Rimskaya-Korsakova,
nadezdarkorsakova@gmail.com

## ABSTRACT

Macrofauna can contribute substantially to the organic matter cycling on the seafloor, yet the role of terrestrial and chemosynthetic organic matter in the diets of microphagous (deposit and suspension) feeders is poorly understood. In the present study, we used stable isotopes of carbon and nitrogen to test the hypothesis that the terrestrial organic matter supplied with river runoff and local chemosynthetic production at methane seeps might be important organic matter sources for macrofaunal consumers on the Laptev Sea shelf. We sampled locations from three habitats with the presumed differences in organic matter supply: "Delta" with terrestrial inputs from the Lena River, "Background" on the northern part of the shelf with pelagic production as the main organic matter source, and "Seep" in the areas with detected methane seepage, where chemosynthetic production might be available. Macrobenthic communities inhabiting each of the habitats were characterized by a distinct isotopic niche, mostly in terms of $\delta^{13}C$ values, directly reflecting differences in the origin of organic matter supply, while $\delta^{15}N$ values mostly reflected the feeding group (surface deposit/suspension feeders, subsurface deposit feeders, and carnivores). We conclude that both terrestrial and chemosynthetic organic matter sources might be substitutes for pelagic primary production in the benthic food webs on the largely oligotrophic Laptev Sea shelf. Furthermore, species-specific differences in the isotopic niches of species belonging to the same feeding group are discussed, as well as the isotopic niches of the symbiotrophic tubeworm *Oligobrachia* sp. and the rissoid gastropod *Frigidoalvania* sp., which are exclusively associated with methane seeps.

## INTRODUCTION

It is a widely accepted paradigm that environment and biodiversity jointly drive ecosystem functioning (*van der Plas, 2019*). Specifically, accounting for the functional diversity of benthic biota might improve understanding of various ecosystem functions on the seafloor including organic matter cycling (*Snelgrove et al., 2018*). Within marine sediments, organic matter mineralization in part depends on the processing of organic matter by macrofauna (*Woulds et al., 2016*; *Middelburg, 2018*). Different macrobenthic communities respond differently to inputs of organic matter depending on species and functional composition (*Josefson, Forbes & Rosenberg, 2002*; *Norling et al., 2007*; *Karlson et al., 2010*; *Belley & Snelgrove, 2017*). Moreover, organic matter differs in its bioavailability for macrobenthic consumers as well as its nutritional value depending on its origin and quality (*Mayor et al., 2012*; *Hunter et al., 2013*; *Campanyà-Llovet, Snelgrove & Parrish, 2017*). Therefore, identifying the organic matter sources for benthic food webs and functional differences within macrobenthic consumers are prerequisites for an improved understanding of benthic organic matter cycling.

There are several major sources of organic matter on the Arctic shelf: phytoplankton and ice algae are the main sources of marine primary production (*Sakshaug, 2004*; *Søreide et al., 2006*), while river runoff and coastal erosion supply organic matter of terrestrial origin (*Goñi et al., 2000*; *Knies & Martinez, 2009*; *Vonk et al., 2012*; *Xiao, Fahl & Stein, 2013*). Macrobenthic communities mainly rely on the export flux of organic matter from the euphotic zone, both of sympagic (ice algae) and pelagic (phytoplankton) origin (*Søreide et al., 2013*; *Mäkelä, Witte & Archambault, 2017*). There is also evidence of terrestrial organic matter being incorporated into benthic food webs, although it is considered more refractory compared to organic matter of marine origin (*Dunton, Schonberg & Cooper, 2012*; *Bell, Bluhm & Iken, 2016*; *Harris et al., 2018*). The exact quality and quantity of organic matter reaching the seafloor might depend on many factors and processes, both biotic and abiotic, including depth, water mass properties and currents, overall primary production, terrestrial inputs, grazing by zooplankton, and microbial degradation of sinking organic matter (*Iken, Bluhm & Gradinger, 2005*; *Roy et al., 2015*; *Stasko et al., 2018*). Locally, benthic primary production is important for benthic food webs, including microphytobenthos and macroalgae in the coastal habitats (*Woelfel et al., 2010*, *Renaud et al., 2015*) and chemosynthetic production in the areas with seabed emission of reduced compounds, such as methane and hydrogen sulfide, also known as "methane seeps" and "hydrothermal vents" (*Sweetman et al., 2013*; *Åström, Bluhm & Rasmussen, 2022*). Consequently, regional and local ecosystem features might drive differences in organic matter availability and consumption by macrofauna. In this respect, the Laptev Sea appears to be among the least studied Arctic seas (*Kędra et al., 2015*). This part of the Arctic is considered a region of high biogeochemical activity due to considerable river runoff, coastal erosion, and methane seepage (*Dittmar & Kattner, 2003*, *Vonk et al., 2012*, *Savvichev et al., 2018*, *Baranov et al., 2020*).

Macrofauna can be generally divided into microphages (*e.g.*, deposit and suspension feeders) and macrophages (*e.g.*, carnivores and herbivores), but some taxa, such as

siboglinid tubeworms, are symbiotrophic and host chemosynthetic bacteria (*Jumars, Dorgan & Lindsay, 2015*; *Åström et al., 2020*). Soft-bottom macrobenthic communities are typically dominated by deposit and suspension feeding fauna (*Kokarev et al., 2017*; *Kokarev et al., 2021a*). The food items consumed by microphages include suspended and freshly deposited plankton, such as diatoms and flagellates, but also organic matter of various origins in the form of particulate matter, or detritus, and associated microorganisms (*Levinton, 1972*; *North et al., 2014*; *Kędra et al., 2021*). Since the visual analysis of food resources utilized by microphagous feeders can be problematic due to the small size of food items and high amount of consumed inorganic particles, indirect methods, such as stable isotopes of carbon and nitrogen, proved to be useful in identifying sources of organic matter consumed by macrofauna (*Yokoyama & Ishihi, 2007*; *North et al., 2014*). The ratio of $^{15}$N to $^{14}$N ($\delta^{15}$N) is reflective of trophic level due to enrichment at each trophic level by 2.5–5‰ while the ratio of $^{13}$C to $^{12}$C ($\delta^{13}$C) changes little in the food chains and is therefore used to trace carbon sources within food webs (*Bearhop et al., 2004*). Stable isotopes can be used both for comparing resource use between species (*i.e.*, isotopic niche overlap) or community-wide aspects of trophic structure or resource use (*Karlson, Gorokhova & Elmgren, 2015*; *Swanson et al., 2015*; *Jackson et al., 2011*; *Włodarska-Kowalczuk et al., 2019*). However, in order to differentiate the consumption of different carbon sources, these carbon sources must differ in their isotopic signature. For instance, on the Laptev Sea shelf, macrobenthic consumers can be expected to exploit terrestrial, marine, and chemosynthetic organic matter (*Kokarev et al., 2017*; *Vedenin et al., 2020*). Carbon isotopic values of terrestrial organic matter are in the range of −28 to −26‰, while phytoplankton-derived organic matter is typically in the range of −22 to −20‰, although marine signatures can be more variable in the Arctic, especially when ice algae are considered with carbon isotopic signature in the range of −15 to −8‰ (*Stein & Macdonald, 2004*). However, there are seasonal and geographic variation in the baseline isotopic signatures of marine and terrestrial organic matter (*Tamelander et al., 2009*; *Søreide et al., 2013*; *Bell, Bluhm & Iken, 2016*; *McClelland et al., 2016*). In the areas of methane seepage, distinct microbial communities in the sediments are formed that include chemosynthetic microorganisms capable of utilizing either methane (methanotrophs) or hydrogen sulfide (thiotrophs) as an energy source (*Åström et al., 2020*). Carbon in the biomass of sulfur-oxidizing bacteria and methanotrophic microorganisms is usually characterized by very low $\delta^{13}$C values (<−30‰) (*Levin & Michener, 2002*; *Decker & Olu, 2012*; *Åström, Bluhm & Rasmussen, 2022*).

In the present study, we hypothesized that macrofauna on the Laptev Sea shelf might utilize organic matter of three origins: marine pelagic, terrestrial, and chemosynthetic. To test this hypothesis, we have obtained data on the carbon and nitrogen stable isotope composition of macrofauna from three soft-bottom habitats with the presumed differences in organic matter sources. The "Background" habitat on the northern part of the shelf was chosen to represent the baseline for the consumption of pelagic primary production. In the "Delta" habitat reliance on the terrestrial organic matter supplied with Lena River runoff was expected in addition to the marine primary production. Finally, the "Seep" habitat in the areas with detected methane seepage where additional bacterial chemosynthetic

production might be available for consumption, which is absent in the "Background" and "Delta". Specifically, we aimed to answer the following questions: (1) How different are the isotopic niches of the studied communities, and (2) whether pronounced isotopic differences exist between species with similar feeding habits?

## MATERIALS AND METHODS

### Study area and definition of habitats

The Laptev Sea is a Siberian epicontinental sea with an average water depth of around 50 meters (Fig. 1). Prolonged ice coverage and considerable freshwater runoff, mainly supplied by the Lena River, largely shape ecological patterns on the eastern part of the Laptev Sea shelf (*Schmid et al., 2006*). The primary production and the fluxes of phytoplankton-derived organic matter to the seafloor generally decrease with increasing latitude (*Stein & Fahl, 2004*; *Demidov, Sheberstov & Gagarin, 2020*). The highest organic carbon content in the sediments is recorded east of the Lena Delta and is the consequence of the accumulation of terrestrial organic matter supplied with the river runoff (*Stein & Fahl, 2004*; *Xiao, Fahl & Stein, 2013*). There is a pronounced latitudinal gradient in the composition of macrobenthic communities associated with the riverine discharge and regime of sedimentation (*Kokarev et al., 2017*). Close to the Lena Delta, species-poor communities are mainly dominated by the bivalve *Portlandia arctica*, while on the northern part of the shelf, the number of species increases, and tube-dwelling annelid worms are abundant, including *Myriochele heeri*, *Owenia polaris*, and *Maldane sarsi*. In addition, on the northern part of the shelf, specific benthic habitats are associated with the methane seeps (*Baranov et al., 2020*; *Vedenin et al., 2020*). These habitats are characterized by the presence of carbonate crusts and white microbial mats on the sediment surface. Few species are found exclusively at the methane seeps, most notably the siboglinid tubeworm *Oligobrachia* sp. and the rissoid gastropod *Frigidoalvania* sp.

For this study, the sampling stations, nine in total, were divided into three types of soft-bottom benthic habitats based on the presumed differences in organic matter supply to the sediments (Fig. 1): the habitat in the vicinity of the Lena Delta with high sedimentation rates of riverine organic matter ("Delta"), the northern shelf habitat in the area with no detected methane seepage ("Background") and methane seeps ("Seep"). Two stations from the "Delta" habitat, 6976 and 6977, were treated as one location as these stations are located close to each other (<500 m) and no small-scale heterogeneity in terms of organic matter sources was expected in this habitat based on the prior knowledge (*Kokarev et al., 2017*). Contrary, small-scale heterogeneity was expected at the "Seep" habitat (stations 5625, 6939, 6947, 6952, 6953, 6992) as there is evidence that chemosynthetic production is distributed very locally, *e.g.*, patches of white microbial mats (*Baranov et al., 2020*; *Vedenin et al., 2020*). The "Background" station (6950) was located outside of the influence of the Lena River runoff but close to the "Seep" stations so that they would be similar in terms of bentho-pelagic coupling and fluxes of organic matter to the seafloor but differ in the sediment biogeochemical processes associated with microbial activities (*Savvichev et al., 2018*; *Savvichev et al., 2023*). Although the "Background" and "Delta" had only one location sampled, we consider the number of animals sampled from

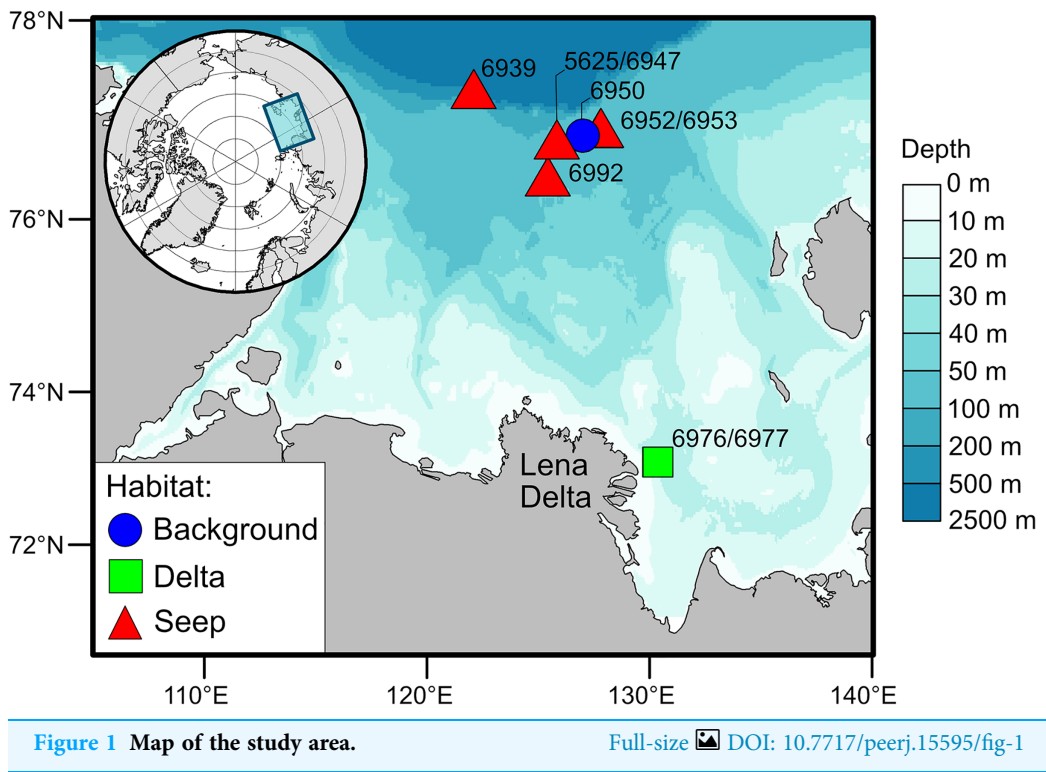

**Figure 1 Map of the study area.**

each of these habitats to be sufficient to represent the isotopic niche of respective communities (Table 1; Table S1).

## Collection and processing of macrofauna

For the present study, most of the material was collected during the 82nd cruise of the R/V "Akademik Mstislav Keldysh" in October 2020 (data on the sampling stations are available in the Table S2). The sediment samples were obtained at eight stations using a grab sampler "Okean" (0.25 m$^2$). Sediments from the grab samples were sieved on the 0.5 mm mesh, and the residues were sorted under a stereomicroscope onboard. Specimens were cleared of debris (and polychaetes removed from their tubes), identified to the species level wherever possible, and frozen at −20 °C before further processing. In addition, the gastropods *Frigidoalvania* sp., collected in September 2017 during the 69th cruise of the R/V "Akademik Mstislav Keldysh" with a Sigsbee trawl and kept frozen at −20 °C, were used in this study (station 5625, for more details on 69th cruise see *Vedenin et al., 2020* and *Baranov et al., 2020*). These gastropods were not found during the 82nd cruise, most probably due to a very patchy distribution based on grab samples (*Vedenin et al., 2020*). Overall, 24 taxa were sampled in this study (Table 1; Figs. 2 and 3). In the lab, the shells of mollusks were removed with a razor blade and tweezers, and for larger specimens the foot muscle was dissected for stable isotope analyses. For the smaller bivalves, annelids, and *Frigidoalvania* sp. all the soft tissues were used. For *Thyasira* cf. *gouldii, Parathyasira dunbari, Yoldiella lenticula, Yoldiella solidula*, and one sample of *Ennucula tenuis*, several individuals of the same species were pooled in one sample due to the very low amount of soft tissues in a single specimen. Subsequently, samples were dried at 60 °C and

**Table 1 List of the macrobenthic taxa sampled.** Numbers indicate the number of samples used for stable isotope analyses. Surface Deposit Feeders, SDF; Suspension Feeders, SF; Subsurface Deposit Feeders, SSDF (for details see section "Classification of the macrofauna based on feeding habits"). An asterisk (*) indicates taxa unassigned to a particular feeding habit Station-wise data and all the measured $\delta^{13}C$, $\delta^{15}N$, and C:N values for each sample are available in the Table S1.

| Taxonomic group | Species | Feeding habit | Habitat | | |
|---|---|---|---|---|---|
| | | | Background | Delta | Seep |
| Bivalvia | *Ennucula tenuis* (Montagu, 1808) | SDF/SF | 1 | | 1 |
| Bivalvia | *Macoma calcarea* (Gmelin, 1791) | SDF/SF | | 2 | 2 |
| Bivalvia | *Macoma moesta* (Deshayes, 1855) | SDF/SF | 1 | 2 | |
| Bivalvia | *Macoma* sp. | SDF/SF | 1 | | 2 |
| Bivalvia | *Macoma torelli* (A. S. Jensen, 1905) | SDF/SF | | | 1 |
| Bivalvia | *Nuculana pernula* (O. F. Müller, 1779) | SDF/SF | 1 | | 12 |
| Bivalvia | *Periploma aleuticum* (A. Krause, 1885) | SDF/SF | 1 | | 1 |
| Bivalvia | *Portlandia arctica* (Gray, 1824) | SDF/SF | 2 | 5 | 6 |
| Bivalvia | *Thracia myopsis* Møller, 1842 | SDF/SF | | | 2 |
| Bivalvia | *Yoldia hyperborea* (A. Gould, 1841) | SDF/SF | | | 1 |
| Bivalvia | *Yoldiella lenticula* (Møller, 1842) | SDF/SF | 3 | | 7 |
| Bivalvia | *Yoldiella solidula* Warén, 1989 | SDF/SF | | | 3 |
| Bivalvia | *Parathyasira dunbari* (Lubinsky, 1976)* | n/a | 1 | | 1 |
| Bivalvia | *Thyasira* cf. *gouldii** | n/a | 1 | 6 | 5 |
| Bivalvia | *Montacuta spitzbergensis* Knipowitsch, 1901* | n/a | | | 1 |
| Gastropoda | *Frigidoalvania* sp.* | n/a | | | 6 |
| Annelida | Nephtyidae gen. sp. | Carnivore | 1 | 3 | 14 |
| Annelida | *Maldane sarsi* Malmgren, 1865 | SSDF | 5 | | 19 |
| Annelida | *Pectinaria (Cistenides) hyperborea* (Malmgren, 1866) | SSDF | 2 | 3 | 8 |
| Annelida | *Sternapsis* sp. | SSDF | 2 | 3 | 12 |
| Annelida | *Myriochele heeri* Malmgren, 1867 | SDF/SF | 1 | | 28 |
| Annelida | *Owenia polaris* Koh, Bhaud & Jirkov, 2003 | SDF/SF | 1 | | 9 |
| Annelida | *Spiochaetopterus typicus* M Sars, 1856 | SDF/SF | | | 2 |
| Annelida | *Oligobrachia* sp. | Symbiotroph | | | 27 |

homogenized with mortar and pestle. Field samplings and experiments were approved by the Ministry of Education and Science of the Russian Federation (the field study approval number DN-09-54/52).

## Collection and processing of sediments

In addition to animal samples, at each station during the 82$^{nd}$ cruise (except for 6976) surface sediments were sampled from the grab prior to sieving with a spoon and immediately frozen at −20 °C. Subsequently, sediment samples were dried at 60 °C. For the stable isotope analyses, dried samples were divided into two replicates. One replicate was acidified with 2M HCl to remove inorganic carbon and was used to determine the $\delta^{13}C$ value. Since acidification alters the $\delta^{15}N$ value, non-acidified samples were used to determine $\delta^{15}N$ (*Silberberger et al., 2021*).

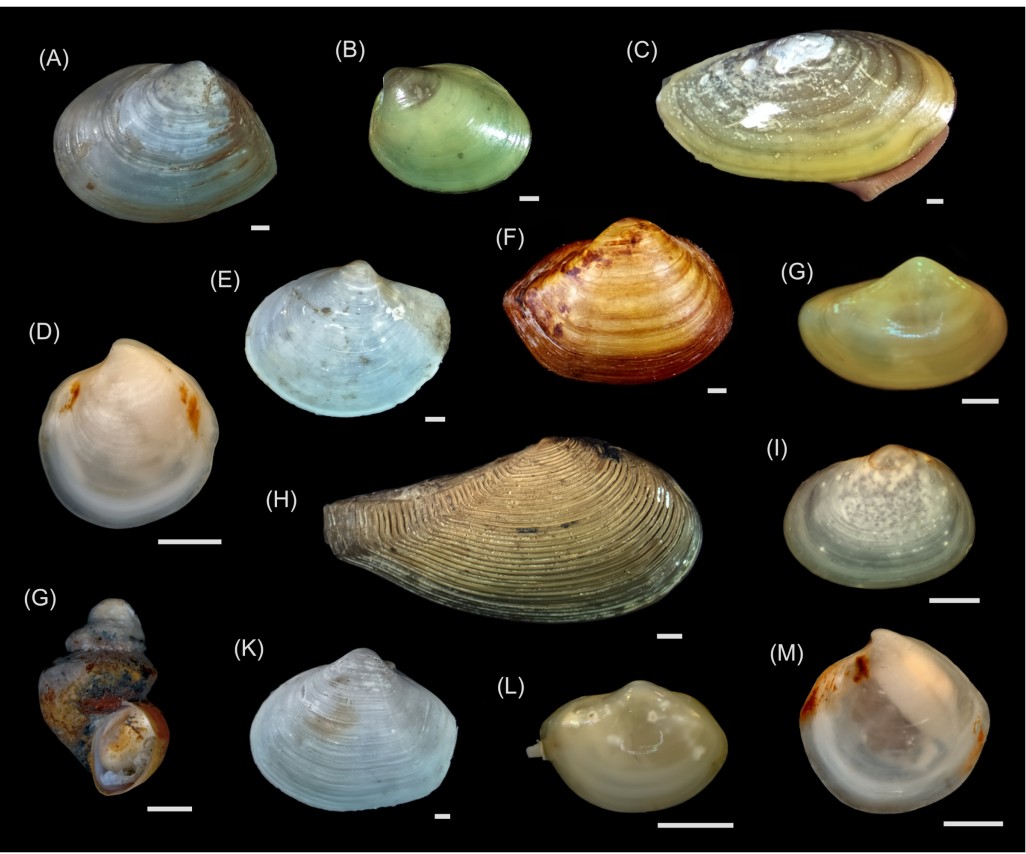

**Figure 2 Macrophotographs of the mollusk species studied.** (A) *Macoma calcarea*, (B) *Ennucula tenuis*, (C) *Yoldia hyperborea*, (D) *Thyasira cf. gouldii*, (E) *Periploma aleuticum*, (F) *Portlandia arctica*, (G) *Yoldiella lenticula*, (H) *Nuculana pernula*, (I) *Montacuta spitzbergensis*, (G) *Frigidoalvania* sp., (K) *Thracia myopsis*, (L) *Yoldiella solidula*, (M) *Parathyasira dunbari*. All scale bars equal 1 mm. Photo credit: Valentin Kokarev.

## Stable isotope analyses

Dried samples (200–500 μg for animal tissues and 3 to 10 mg for sediment samples) were wrapped into tin capsules and analyzed for stable carbon and nitrogen isotope composition using Flash 1112 Elemental Analyzer (Thermo Fisher Scientific, Waltham, MA, USA) and a Thermo Delta V Plus isotope ratio mass spectrometer connected *via* a Conflo IV peripheral at the A.N. Severtsov Institute of Ecology and Evolution RAS. Results are reported in permille (‰) using the delta (δ) notation and were normalized to VPDB and air $N_2$ using USGS40 and USGS41 reference materials (US Geological Survey, Reston, VA, USA). The drift was corrected using an internal laboratory standard (casein). The standard deviation of $\delta^{13}C$ and $\delta^{15}N$ values in the laboratory standard ($n = 8$) was <0.2‰. Although C:N ratios were variable, particularly between foot *vs.* whole clam samples, which is clearly a consequence of differences in lipid content (*Weems et al., 2012*), the data were not corrected for lipids using mathematical corrections as available equations might not be suitable for benthic invertebrates such as bivalves and polychaetes (*Silberberger et al., 2021*). All the measured $\delta^{13}C$, $\delta^{15}N$, and C:N values for each animal sample are available in the Table S1. The values in the text are presented as mean ± SD.

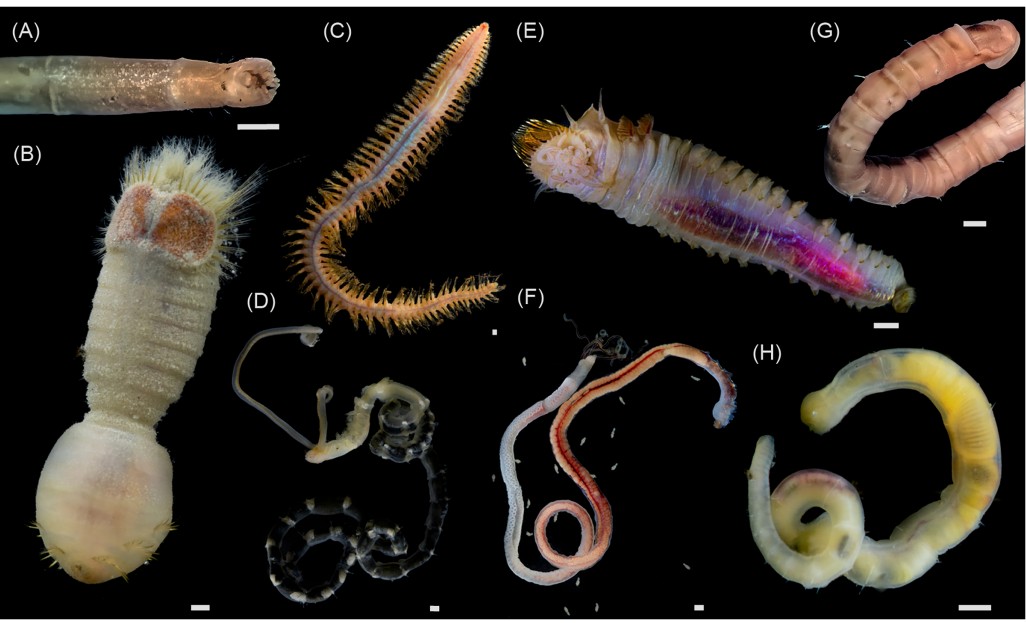

**Figure 3 Macrophotographs of the studied annelids.** All alive except (A) and (G). (A) *Owenia polaris*, (B) *Sternapsis* sp., (C) *Aglaophamus malmgreni* (Nephtyidae), (D) *Spiochaetopterus typicus*, (E) *Pectinaria hyperborea*, (F) *Oligobrachia* sp. (with metatrochophore larvae), (G) *Maldane sarsi*, (H) *Myriochele heeri*. All scale bars equal 1 mm. Photo credit: (A, G) Valentin Kokarev, (B, D–F, H) Petr Kuznetsov, (C) Tina Molodtsova.

## Classification of the macrofauna based on feeding habits

Prior to the data analyses, species were assigned to trophic groups based on *Jumars, Dorgan & Lindsay (2015)* and *Kokarev et al. (2017)* (Table 1). Surface deposit feeders and suspension feeders were combined for two reasons: (1) some species can switch from deposit feeding to suspension feeding, *e.g.*, *Owenia*, *Spiochaetopterus*, *Macoma* (*Jumars, Dorgan & Lindsay, 2015*; *Reid & Reid, 1969*) and (2) both groups utilize similar food sources: fresh phytoplankton material, food resources deposited in the upper layer of the sediments, which are available to suspension feeders through resuspension (*Christensen & Kanneworff, 1985*; *Sokołowski et al., 2014*). Moreover, only two species could be classified as strict suspension feeders: bivalves *Thracia myopsis* and *Periploma aleuticum* (*Sartori & Domaneschi, 2005*; *Morton, 1981*). Subsurface deposit feeders are represented by annelids that feed in the deeper layers of the sediment and rely on more microbially degraded organic matter, which is reflected in higher $\delta^{15}$N values (*North et al., 2014*; *Jumars, Dorgan & Lindsay, 2015*). The symbiotrophic tubeworm *Oligobrachia* sp. and carnivores nephtyid polychaetes were the only taxa in their respective feeding group. In the section "Seep-associated fauna" we used additional data on *Oligobrachia* and $\delta^{13}$C values of methane from North Atlantic and Arctic seeps for comparison on a larger scale (*Gebruk et al., 2003*; *Decker & Olu, 2012*; *Lee et al., 2019a*; *Åström, Bluhm & Rasmussen, 2022*; *Kravchishina et al., 2021*). Several taxa were not assigned to a particular feeding habit: the gastropod *Frigidoalvania* sp., the bivalve *Montacuta spitzbergensis*, and thyasirid bivalves. *Frigidoalvania* sp. is currently known only from the seeps in the Laptev Sea, and little is

known about its feeding habits (*Nekhaev & Krol, 2020*). *M. spitzbergensis* belongs to the family Montacutidae, which comprises commensal bivalves, but its host is unknown with all the specimens (including this study) found freely in sediments (*Kamenev, 2008*). Feeding strategies in the thyasirid bivalves are diverse, particularly in the cryptic *Thyasira* cf. *gouldii*, which might derive its nutrition from sulfide-oxidizing bacteria in the gills, subsurface pedal feeding, and suspension feeding (*Zanzerl & Dufour, 2017*; *Zanzerl et al., 2019*).

## Data analysis

Stable isotope data were visualized as biplots using R version 4.2.2 (*R Core Team, 2022*) and package ggplot2 version 3.4.1 (*Wickham, 2016*). Since the data partially were not normally distributed in bivariate isotopic space (tested with Shapiro-Wilk multivariate normality test in the package RVAideMemoire version 0.9-81-2; *Hervé, 2022*), the data analysis was performed with non-parametric approaches that do not assume normal distribution or homogeneity of variance. Differences between pre-defined groups ("Habitat", "Feeding habit" and "Station") were tested with a two-way non-parametric Scheirer Ray Hare test in the package rcompanion version 2.4.21 (*Mangiafico, 2023*) or one-way Kruskall-Wallis test and subsequent pairwise Dunn's test with Holm's procedure to adjust $p$-value for multiple comparisons in the package dunn.test version 1.3.5. (*Dinno, 2017*). The multivariate differences on the species level were investigated using non-parametric rank-based Wilks' Lambda type statistic, for which $p$-value was calculated using $F$ approximation, and the empirical nonparametric relative treatment effects for each variable in the package npmv version 2.4.0 (*Ellis et al., 2017*). The relative effects quantify the probability that a randomly chosen individual from one species exhibits a higher value for the variable ($\delta^{13}C$ or $\delta^{15}N$) than a randomly chosen individual across all species.

# RESULTS

## Habitat-associated patterns

The sediment stable isotope data showed some differences between "Background", "Seep" and "Delta" habitats: lower $\delta^{13}C$ values were observed for the "Seep" and the "Delta" as well as a considerably lower $\delta^{15}N$ value for "Delta" sediments (Table 2). "Background" and "Seep" habitats had similar $\delta^{15}N$ values.

The patterns within macrobenthic consumers were more prominent. There was a considerable difference between the symbiotrophic annelid *Oligobrachia* sp. from "Seep" stations and all the other macrobenthic taxa due to non-overlapping $\delta^{13}C$ values ($-48.4 \pm 3.1$‰ for *Oligobrachia* sp. and $-24.3 \pm 2.7$‰ for all the other taxa). The dispersion of the *Oligobrachia* sp. samples in the isotopic space was of comparable magnitude to all the other taxa combined (Fig. S1). Therefore, subsequent comparisons of different habitats were performed excluding *Oligobrachia* sp.

The exclusion of *Oligobrachia* sp. revealed the significant differences between macrobenthic consumers from the three habitats (Fig. 4A). The "Delta" community was characterized by lower values of $\delta^{13}C$ ($-27.9 \pm 1.6$‰) compared to the "Background" community ($-22.7 \pm 1.2$‰). The "Seep" community ($-24.0 \pm 2.6$‰) with the widest

**Table 2 $\delta^{13}$C and $\delta^{15}$N measured for the sediment samples.** An asterisk (*) indicates average values for two samples.

| | Background | Delta | Seep | | | | |
|---|---|---|---|---|---|---|---|
| | 6950 | 6977 | 6939 | 6947* | 6952 | 6953 | 6992 |
| $\delta^{13}$C, ‰ | −23.5 | −25.8 | −26.0 | −24.2 | −24.4 | −24.4 | −23.6 |
| $\delta^{15}$N, ‰ | 5.9 | 3.0 | 5.3 | 5.6 | 6.0 | 5.9 | 5.7 |

isotopic niche overlapped both with the "Delta" and "Background" communities. The two-way Scheirer Ray Hare test (excluding taxa unassigned to feeding group and carnivores due to their low representation ($n < 5$) in the "Delta" and "Background") revealed that the effect of the interaction of "Habitat" and "Feeding habit" on the $\delta^{13}$C is not significant (H = 5.2, df = 2, $p$ = 0.075), as well as the main effect of "Feeding habit" (H = 3.6, df = 1, $p$ = 0.056), while the main effect of "Habitat" was significant (H = 31.7, df = 2, $p$ < 0.001). Further, differences in the range of $\delta^{13}$C values were revealed between different "Seep" stations (Fig. 5): while stations 6939 and 6992 were completely within the "Background" range, considerably lower $\delta^{13}$C values were recorded for the stations 6952, 6953, and 6947, with stations 6953 and 6947 being significantly different from "Background" (results of Kruskal-Wallis test and subsequent Dunn's test are presented in the Table S3).

Macrobenthic consumers with different feeding habits could be differentiated based on their $\delta^{15}$N values (Fig. 4B): surface deposit feeders and suspension feeders had the lowest and carnivores the highest $\delta^{15}$N values (7.1 ± 0.9‰ and 11.6 ± 0.6‰, respectively) with subsurface deposit feeders in between (10.3 ± 1.2‰). In addition, centroid of the "Delta" community was clearly skewed towards lower $\delta^{15}$N (7.1 ± 2.2‰) compared to "Background" (8.6 ± 2.3‰) and "Seep" (8.5 ± 2.0‰). The two-way Scheirer Ray Hare test (excluding taxa unassigned to feeding group and carnivores due to their low representation ($n < 5$) in the "Delta" and "Background") revealed that the effect of interaction of "Habitat" and "Feeding habit" on the $\delta^{15}$N is not significant (H = 1.4, df = 2, $p$ = 0.500), while the main effects of "Habitat" (H = 13.5, df = 2, $p$ = 0.001) and "Feeding habit" (H = 93.7, df = 1, $p$ < 0.001) are significant. Among taxa with unassigned feeding habits, thyasirid bivalves and the gastropod *Frigidoalvania* sp. clearly grouped with surface deposit and suspension feeders (Fig. 4B). The single sample of the bivalve *Montacuta spitzbergensis* grouped with a few outliers from the "Seep" community with the lowest $\delta^{13}$C values.

In the following section "Background" and "Seep" samples for surface deposit/suspension feeders and subsurface deposit feeders are discussed together since samples from these two habitats clearly overlap (Figs. 4 and 5).

## Species isotopic niches

### Surface deposit/suspension feeders

"Delta" surface deposit/suspension feeders were represented only by *Portlandia arctica* ($\delta^{13}$C = −29.0 ± 1.0‰; $\delta^{15}$N = 5.6 ± 0.2‰) and *Macoma* spp. ($\delta^{13}$C = −29.0 ± 0.4‰;

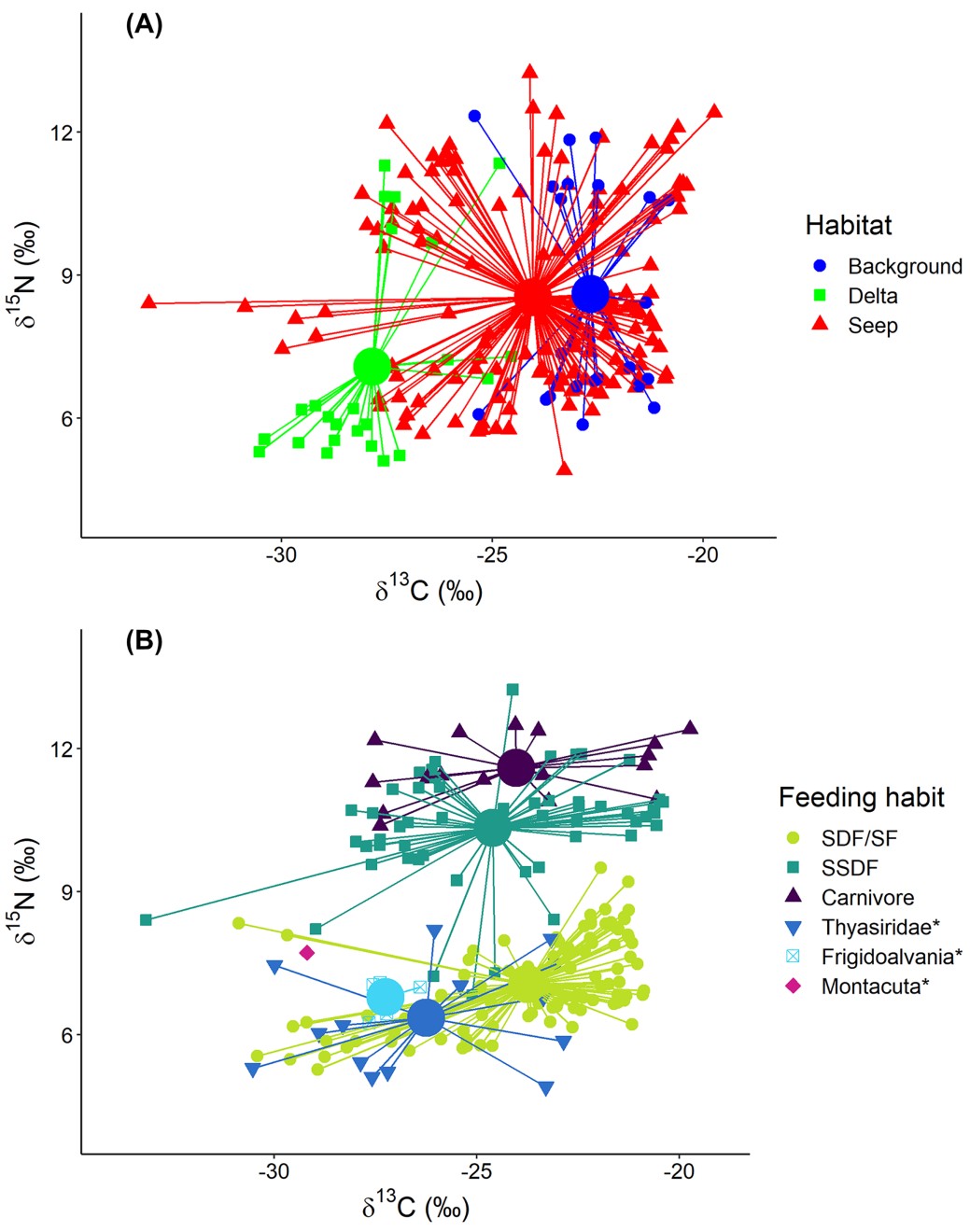

**Figure 4 Stable isotope biplot of δ13C and δ15N values for the macrobenthic taxa excluding** *Oligobrachia* **sp.** Habitats are locations on the Laptev Sea shelf with the presumed differences in organic matter sources for macrofauna (marine *vs.* terrestrial *vs.* chemosynthetic). Group centroids (calculated as mean $\delta^{13}C$ and $\delta^{15}N$ values) are shown. (A) and (B) are the same plot but differently color-coded. Surface Deposit Feeders, SDF; Suspension Feeders, SF; Subsurface Deposit Feeders, SSDF. An asterisk (*) indicates taxa unassigned to a particular feeding habit. Thyasridae was represented by two species: *Thyasira gouldi* and *Parathyasira dunbari*.

$\delta^{15}N = 6.0 \pm 0.3$ ‰). This group occupied a narrow and distinct isotopic niche that did not overlap with the isotopic niches of the same taxa from "Seep" and "Background" (*P. arctica*: $\delta^{13}C = -22.5 \pm 1.7$‰; $\delta^{15}N = 6.5 \pm 0.4$‰; *Macoma* spp.: $\delta^{13}C = -23.5 \pm 1.4$‰; $\delta^{15}N = 6.9 \pm 0.4$‰).

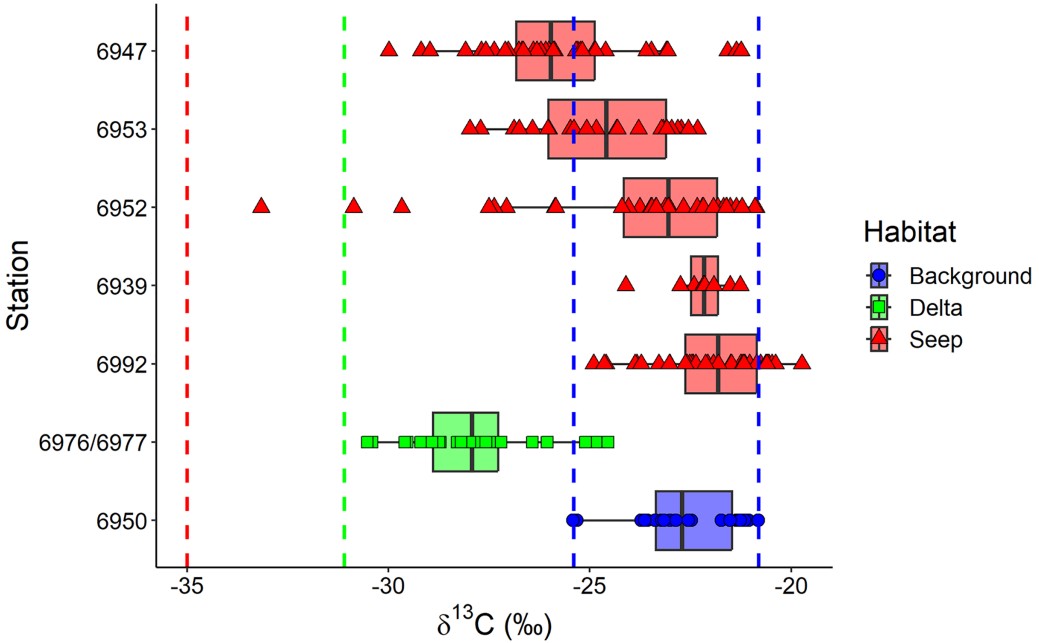

**Figure 5 Boxplot of δ$^{13}$C values for the macrobenthic taxa (excluding *Oligobrachia* sp) grouped by sampling station.** The station 5625, where only *Frigidoalvania* sp. was sampled, is not shown. Blue dashed lines correspond to δ$^{13}$C range (min-max) of the "Background" habitat. The red dashed line corresponds to the carbon isotopic signature (−35) of sulphur-oxidizing bacteria at high-latitude seeps (*Åström, Bluhm & Rasmussen, 2022*). The green dashed line corresponds to the isotopic signature of Lena River (−31.1) particulate organic matter (*McClelland et al., 2016*).

Surface deposit/suspension feeders from "Background" and "Seep" habitats were represented by 12 taxa, and there was a general trend in lower δ$^{13}$C values being associated with lower δ$^{15}$N values (Fig. 6). Two "Seep" samples of the bivalve *Thracia myopsis* had the lowest δ$^{13}$C values (≈−30‰). Oweniid annelids *Myriochele heeri* and *Owenia polaris* along with annelid *Spiochaetopterus typicus* and bivalve *Periploma aleuticum* had values of the δ$^{13}$C higher than −24‰. Contrary, the majority of the other bivalve species (*Yoldiella lenticula*, *Nuculana pernula*, *Portlandia arctica*, *Macoma* spp.) showed a wide range of δ$^{13}$C values with some of the "Seep" samples depleted in $^{13}$C. The bivalve *Y. solidula* had higher δ$^{15}$N values compared to *Y. lenticula*, *N. pernula*, *P. arctica*, and *Macoma* spp. The lowest δ$^{13}$C value (−25.3‰) from the "Background" was observed for the bivalve *Ennucula tenuis*, which appeared to be an outlier compared to the rest of the "Background" species, but a similar value for this species was recorded from the "Seep" habitat. However, the samples of *E. tenuis* had a high C:N ratio (>6), which might have resulted in decreased δ$^{13}$C values due to high lipid content.

The multivariate differences among surface deposit and suspension feeding species from "Background" and "Seep" habitats with sample size ≥7 were significant (Wilks Lambda = 10.3, df1 = 10, df2 = 140, $p < 0.001$). The relative effects revealed that these differences were mainly due to the difference between surface deposit feeding bivalves and

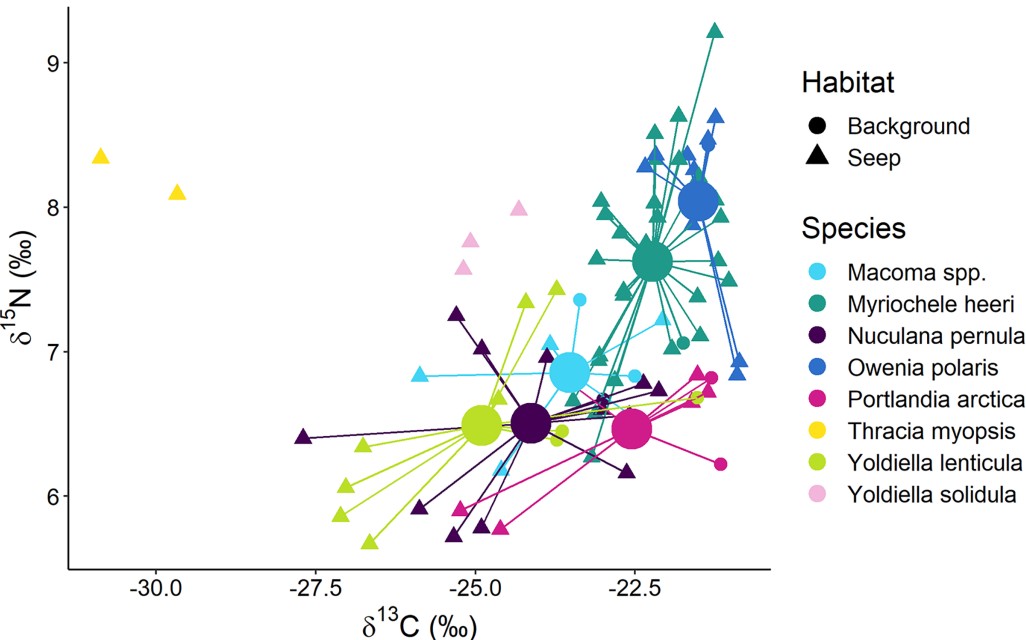

**Figure 6 Stable isotope biplot of δ¹³C and δ¹⁵N values for surface deposit and suspension feeders from the northern part of the Laptev Sea shelf.** Group centroids (calculated as mean δ¹³C and δ¹⁵N values) are shown for groups with 7 or more samples.The following samples are not shown to improve readability: "Background" *E. tenuis* (δ¹³C = −25.3‰, δ¹⁵N = 6.1‰), "Seep" *E. tenuis* (δ¹³C = −25.2‰, δ¹⁵N = 5.8‰), "Background" *P. aleuticum* (δ¹³C = −22.6‰, δ¹⁵N = 6.8‰), "Seep" *P. aleuticum* (δ¹³C = −21.9‰, δ¹⁵N = 6.5‰), "Seep" *S. typicus* (δ¹³C = −22.8‰, δ¹⁵N = 7.7‰ and δ¹³C = −21.3‰, δ¹⁵N = 8.1‰), "Seep" *Y. hyperborea* (δ¹³C = −24.9‰, δ¹⁵N = 6.4‰).

oweniid polychaetes, both in terms of δ¹³C and δ¹⁵N (Table 3). Moreover, *O. polaris* had the most distinct niche.

### Subsurface deposit feeders

The polychaetes *Sternapsis* sp. and *Pectinaria hyperborea* from the "Delta" habitat had very different δ¹⁵N values (6.8–7.3‰ for *Sternapsis* sp. and 9.7–10.7‰ for *P. hyperborea*), and therefore their isotopic niches did not overlap. Within "Background" and "Seep" habitats three sampled species had three overlapping but significantly different isotopic niches (Fig. 7; Wilks Lambda = 10.2, df1 = 4, df2 = 88, *p* < 0.001). The relative effects revealed that *M.sarsi* had generally higher δ¹⁵N, while *Sternapsis* sp. had the highest δ¹³C value (Table 4).

### Carnivores

Represented by the single taxon (nephtyid polychaetes), this group showed the highest δ¹⁵N (11.6 ± 0.6‰) values but had a similar δ¹³C (24.0 ± 2.7‰) range to the surface deposit/suspension feeders and subsurface deposit feeders (Fig. 4B).

### Thyasiridae

Thyasirid bivalves grouped with suspension/surface deposit feeders (Fig. 4B). However, two samples of *Parathyasira dunbari* could be distinguished from *Thyasira* cf. *gouldii* on
**Table 3 Nonparametric relative effects for surface deposit and suspension feeders.** The relative effects quantify probability that a randomly chosen individual from one species exhibits a higher value for the variable ($\delta^{13}$C or $\delta^{15}$N) than a randomly chosen individual across all species.

| Species | $\delta^{13}$C | $\delta^{15}$N |
|---|---|---|
| *Macoma* spp. | 0.35714 | 0.41466 |
| *Myriochele heeri* | 0.60099 | 0.66906 |
| *Nuculana pernula* | 0.28921 | 0.27123 |
| *Owenia polaris* | 0.79805 | 0.80195 |
| *Portlandia arctica* | 0.59578 | 0.25731 |
| *Yoldiella lenticula* | 0.20649 | 0.25909 |

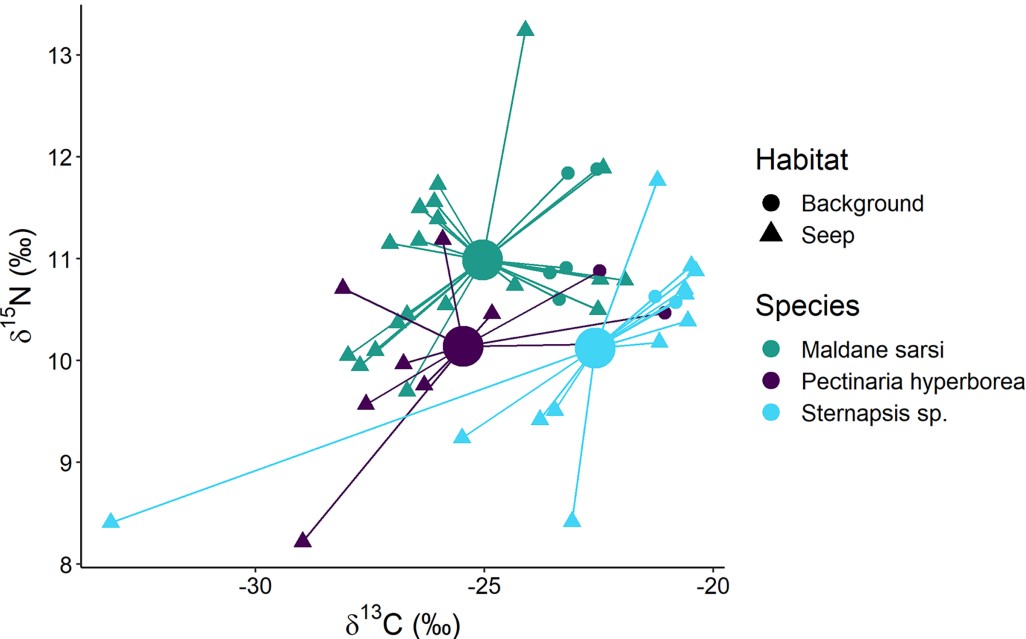

**Figure 7 Stable isotope biplot of $\delta^{13}$C and $\delta^{15}$N values for subsurface deposit feeders from the northern part of the Laptev Sea shelf.** Group centroids (calculated as mean $\delta^{13}$C and $\delta^{15}$N values) are shown.

**Table 4 Nonparametric relative effects for subsurface deposit.** The relative effects quantify probability that a randomly chosen individual from one species exhibits a higher value for the variable ($\delta^{13}$C or $\delta^{15}$N) than a randomly chosen individual across all species.

| Species | $\delta^{13}$C | $\delta^{15}$N |
|---|---|---|
| *Maldane sarsi* | 0.41667 | 0.62153 |
| *Pectinaria hyperborea* | 0.37708 | 0.36771 |
| *Sternapsis* sp. | 0.73065 | 0.38616 |

the northern part of the shelf by lower $\delta^{15}N$ values (4.9‰ and 5.9‰ *vs.* 7.5 ± 0.5‰). Moreover, *P. dunbari* had the lowest recorded $\delta^{15}N$ value for "Background" and "Seep" habitats among all the sampled taxa, excluding *Oligobrachia* sp.

### Seep-associated fauna

Two species were associated exclusively with the "Seep" sites: the rissoid gastropod *Frigidoalvania* sp. and the tubeworm *Oligobrachia* sp. *Frigidoalvania* sp. had relatively low values of $\delta^{13}C$ (−27.2 ± 0.5‰) and a distinct niche among surface deposit/suspension feeders from the seep sites (Fig. 4B).

The samples of *Oligobrachia* sp. had a very wide dispersion in the isotopic space and had the lowest values both for $\delta^{13}C$ and $\delta^{15}N$ (down to −53.9‰ and to −4.0‰, respectively; Fig. S1). The analysis of *Oligobrachia* isotopic signatures from different seep locations in the North Atlantic and the Arctic revealed no clear relationship between $\delta^{13}C$ of methane and $\delta^{13}C$ of *Oligobrachia* (Fig. S2).

## DISCUSSION

### Habitat-associated patterns

Macrobenthic communities from the three studied habitats differed in terms of the $\delta^{13}C$ range, reflecting variability in organic matter sources for macrobenthic communities. However, there were no differences in the $\delta^{13}C$ range between different feeding groups, including primary consumers (deposit and suspension feeders) and secondary consumers (carnivores). The differences in the $\delta^{13}C$ range between habitats were much more pronounced for macrofauna than for sediments, thus, the isotopic signatures of the sediments cannot be used as a food source baseline. "Background" macrofaunal samples had $\delta^{13}C$ values larger than −24‰ (except for two samples with values ≈−25‰), indicating the predominant reliance of the community on marine organic matter (*Vonk et al., 2012*; *Bell, Bluhm & Iken, 2016*). However, the role of marine organic matter decreased for "Delta" and partially for "Seep" habitats.

### Delta

On the eastern Laptev Sea and East Siberian Sea shelf macrobenthic communities are influenced by riverine input and regime of sedimentation, which results in low species and functional diversity close to the deltas due to environmental filtering (*Kokarev et al., 2017*, *2021a*). The species found close to the Lena Delta, are wide-spread shelf species, which can successfully establish a population in waters with warmer temperatures, fluctuating salinity, and enhanced sedimentation (*e.g.*, *Portlandia arctica*, *Macoma* spp., *Thyasira* cf. *gouldii*, *Sternapsis* sp., *Pectinaria hyperborea*, nephtyid annelid *Aglaophamus malmgreni*). In the present study, we showed that these species might consume organic matter of different origins depending on availability as inferred from their variable $\delta^{13}C$ signatures. Macrofauna from the "Delta" habitat, especially surface deposit feeding bivalves, was considerably depleted in $^{13}C$ and slightly depleted in $^{15}N$ relative to "Background" (down to −30.5‰ and 5.3‰, respectively) directly reflecting the isotopic signature of Lena River particulate organic matter ($\delta^{13}C$ = −31.1 ± 0.8‰; *McClelland et al., 2016*) and terrestrial end-member values used for the Arctic shelf ($\delta^{13}C$ = −28.8 ± 3.2‰, $\delta^{15}N$ = 0.8 ± 1.0‰;
*Bell, Bluhm & Iken, 2016*). This fact suggests that terrestrial organic matter is a primary food source for the "Delta" community at least during the season the samples were taken. Such low values for $\delta^{13}$C and $\delta^{15}$N were not previously recorded for the Arctic macrofauna in the areas influenced by river runoff (*Dunton, Schonberg & Cooper, 2012*; *Bell, Bluhm & Iken, 2016*; *Harris et al., 2018*; *Stasko et al., 2018*), which either reflects the higher contribution of terrestrial organic matter to the benthic food web close to the Lena Delta, or seasonal/geographical variations in the isotopic signatures of fluvial particulate organic matter (*McClelland et al., 2016*).

### Seep

Little is known about the nutrition of macrobenthos, particularly deposit feeders, from the arctic cold seeps (*Åström et al., 2020*). Our data revealed a wider $\delta^{13}$C range for the "Seep" community relative to "Background" indicating that carbon of chemosynthetic origin is actively consumed by the non-symbiotrophic deposit feeding macrofauna in addition to photosynthetic carbon. Individuals of many species were more depleted in $^{13}$C at seep sites compared to "Background", which is consistent with patterns observed at some other methane seeps, usually located below shelf depths (*Levin & Michener, 2002*; *Levin, 2005*; *Thurber et al., 2010*; *Sellanes et al., 2011*), and indicates partial reliance on chemosynthetic production which is $^{13}$C-depleted ($\delta^{13}$C $\approx -35$‰ for sulfur-oxidising bacteria from white microbial mats at high-latitude seeps, *Åström et al., 2020*). It should be noted, however, that for many species from "Seep" sites along with more depleted isotopic signatures, isotopic signatures within the "Background" range were observed. Moreover, samples from stations 6939 and 6992 were completely within "Background" $\delta^{13}$C range. Both of these facts suggest the patchy distribution of chemosynthetic organic matter and its opportunistic consumption. Indeed, video observations from the seeps in the Laptev Sea documented patchy distributions of white microbial mats at scales <1 m (*Baranov et al., 2020*). However, these results differ from the shelf of the Barents Sea, where differences between isotopic niches of seep and non-seep communities were less pronounced with no apparent consumption of chemosynthetic organic matter at shallow (<200 m) seeps (*Åström, Bluhm & Rasmussen, 2022*). This supports the previous hypothesis that in the less productive Laptev Sea chemosynthetic organic matter is more readily consumed by macrofauna supporting higher abundances at the seep sites (*Vedenin et al., 2020*), which likely makes the Laptev Sea seeps more of an exception among the shallow-water seeps (*Dando, 2010*).

## Species isotopic niches

### Surface deposit and suspension feeders

Surface deposit feeders from the "Delta" habitat, represented by bivalves *Macoma* spp. and *Portlandia arctica*, had very similar isotopic signatures. On the northern part of the shelf at "Seep" and "Background" habitats, similarly, deposit feeding bivalves *Macoma* spp., *P. arctica*, *Yoldiella lenticula*, and *Nuculana pernula* had high niche overlap and appeared to have a contribution from chemosynthetic carbon to their diet at "Seep" sites. This finding corroborates previous results that different species of surface deposit feeding

bivalves generally consume similar food sources from the sediment pool (*North et al., 2014*; *Oxtoby et al., 2016*). Interestingly, *Yoldiella solidula* had slightly higher $\delta^{15}N$ values compared to the above-mentioned bivalves, possibly indicating a different resource use pattern for this species.

Oweniids *Myriochele heeri* and *Owenia polaris* had niches distinct from each other and limited overlap with most abundant deposit feeding bivalves. Oweniids might selectively consume algal material recently deposited on the seafloor or from the near-bottom suspension (*Jumars, Dorgan & Lindsay, 2015*). The high densities of oweniid polychaetes are also sometimes linked to the input of fresh pelagic organic matter to the seafloor (*Fiege, Kröncke & Barnich, 2000*; *Kokarev et al., 2021b*). In our study, *M. heeri* and *O. polaris* had $\delta^{13}C > -24‰$, indicative of phytoplankton consumption (*Vonk et al., 2012*; *Bell, Bluhm & Iken, 2016*). Probably, dense populations of these species on the northern Laptev Sea shelf (*Kokarev et al., 2017*) rely less on bacterially reworked sediment pool of organic matter compared to deposit feeding bivalves and selectively feed on fresh phytodetritus, avoiding consumption of bacterially derived chemosynthetic organic matter at seep sites. The differences in the isotopic niches between *M. heeri* and *O. polaris* are most probably related to morpho-functional differences between the two genera: *Owenia* uses a tentacular crown to feed, while *Myriochele* lacks any anterior appendages (Fig. 3; *Jumars, Dorgan & Lindsay, 2015*).

Consumption of chemosynthetic carbon was not evident also for the polychaete *Spiochaetopterus typicus* and the bivalves *Periploma aleuticum*, *Yoldiella solidula*, and *Ennucula tenuis*, but only a few samples of these species were analyzed. However, two samples of the bivalve *Thracia myopsis* and a single sample of the bivalve *Montacuta spitzbergensis* were depleted in $^{13}C$ characteristic of chemosynthetic carbon consumption. *T. myopsis* is a strict suspension feeder and has common for the genus mucous lining of the siphon passage in the sediment that restricts the deposited sediment from entering the inhalant siphon aperture (*Sartori & Domaneschi, 2005*). Thus, *T. myopsis* feeds on suspended organic matter, which was shown to be depleted in $^{13}C$ at the seep sites ($\delta^{13}C = -34.8$ to $-35.9‰$; *Savvichev et al., 2018*).

### Subsurface deposit feeders

Subsurface deposit feeders, *e.g.*, the polychaete *Pectinaria hyperborea*, consume mainly bacterially reworked detritus and are characterized by higher $\delta^{15}N$ relative to surface deposit feeding bivalves (*North et al., 2014*). In the present study, we observed a similar pattern. However, three of the studied species had distinct isotopic niches. In the "Delta" community, *Sternapsis* sp. and *P. hyperborea* consumed different resources, while on the northern part of the shelf, at "Background" and "Seep" sites, their isotopic niches partially overlapped with each other and *Maldane sarsi*. It is not clear how these species partition resources, but possible mechanisms include feeding at different sediment levels or selection for different particle sizes (*Hughes, 1979*; *Whitlatch, 1980*). For all three species consumption of chemosynthetic organic matter was evident at the seep sites.

### Thyasiridae

Thyasirid bivalves grouped with surface deposit feeders, although they form pedal tracts that penetrate deep into the sediment (*Zanzerl & Dufour, 2017*), while specimens identified as *Thyasira* cf. *gouldii* can be also symbiotrophic (*Batstone et al., 2014*). Isotopic signatures of *T.* cf. *gouldii* in the present study varied widely among habitats suggesting direct consumption of deposited organic matter in the surface sediment layer rather than symbiont-based nutrition. Interestingly, samples of *Parathyasira dunbari* with similar $\delta^{13}C$ values to *T.* cf. *gouldii* had lower $\delta^{15}N$ values. Such a pattern was also observed in Bonne Bay, Newfoundland, where symbiotic *T.* cf. *gouldii*, asymbiotic *T.* cf. *gouldii*, and *Parathyasira* sp. could be distinguished by $\delta^{15}N$ but not $\delta^{13}C$ values, and in conjunction with fatty acid analysis it was hypothesized that *Parathyasira* sp. might partially rely on free-living sulfur-oxidizing bacteria in the sediment (*Zanzerl et al., 2019*). We observed numerous bacteria on the inner fold of the mantle margin for *Parathyasira dunbari* during SEM investigation of this species, but not for *T.* cf. *gouldii*, supporting this hypothesis (Kokarev et al., in prep.).

### Seep-associated fauna

Rissoid gastropods are characteristic of seep and vent fauna in the Arctic and North Atlantic, where they are associated with bacterial mats of sulfur-oxidizing bacteria, particularly of the genus *Sulfurovum* (*Gebruk et al., 2003*, *Decker & Olu, 2012*; *Sweetman et al., 2013*; *Sen et al., 2019*). *Sulfurovum* is abundant also in seep bacterial communities in the Laptev Sea (*Savvichev et al., 2018*). Although in our study rissoid gastropod *Frigidoalvania* sp. showed quite low $\delta^{13}C$ values ($-27.2\% \pm 0.5‰$) relative to the "Background" surface deposit feeders, these values were quite high relative to the ones measured for *Alvania* sp. from the seeps in the North Atlantic (*Decker & Olu, 2012*). This indicates either a feeding habit on a mixture of different bacteria (*Sweetman et al., 2013*) or probably a mixed diet with significant contributions of photosynthetic organic matter.

The siboglinid worm *Oligobrachia* sp. showed very variable isotopic signatures in the Laptev Sea and across a wider geographical range, which is not directly related to the isotopic signature of methane (Fig. S2). Although there are several cryptic *Oligobrachia* species in the Arctic and North Atlantic, similar symbiotic sulfur-oxidizing bacteria were detected in all the species (*Lösekann et al., 2008*; *Sen et al., 2018*; *Lee et al., 2019b*; *Karaseva et al., 2021*). A variable pattern in the isotopic signatures might be a result of autotrophic fixation of inorganic carbon from a mixture of sources: the inorganic carbon from the overlaying seawater and a more $^{13}C$-depleted pore-water inorganic carbon resulting from anaerobic oxidation of methane in the sediment (*Lösekann et al., 2008*; *Lee et al., 2019a*). This contrasts with the pattern observed for the siboglinid *Siboglinum poseidoni* hosting methanotrophic bacteria, which carbon isotopic signature directly reflects the isotopic signature of methane (*Schmaljohann et al., 1990*). High variation in $\delta^{15}N$ values might also indicate a wide spectrum of nitrogen sources available for *Oligobrachia*, possibly including local assimilation of inorganic nitrogen (*Levin, 2005*) or amino acids from dissolved organic matter (*Southward & Southward, 1981*). Therefore, *Oligobrachia* might benefit from a wide spectrum of carbon and nitrogen sources available for the worm

simultaneously, resulting in high variability of isotopic signatures even on small scales. More studies are needed on the biology of this species and its symbiotic bacteria in order to assess its role in carbon and nutrient cycling at methane seeps.

## CONCLUSION

Our study revealed that the terrestrial and chemosynthetic organic matter might be more substantial food sources for macrofauna than previously demonstrated for the Arctic shelf. Distribution of many species, *e.g.*, surface deposit feeding bivalves and subsurface deposit feeding polychaetes found in all three habitats, is not constrained by the origin of organic matter as demonstrated by the high variability of their isotopic signatures and suggests opportunistic consumption of available resources. However, differences in resource use among some species with the same feeding habits were observed, particularly between oweniid polychaetes and bivalves, suggesting that species-specific traits might be important for ecosystem functioning (*Norling et al., 2007*; *Godbold, Rosenberg & Solan, 2009*). These findings might have important implications for assessing macrobenthic carbon demand and the carbon budget of the Arctic shelf ecosystem (*Schmid et al., 2006*; *Zaborska et al., 2018*).

It must be noted, however, that both consumptions of terrestrial and chemosynthetic organic matter results in lower $\delta^{13}C$ values of macrofaunal consumers, making the interpretation of data contextual. Therefore, a combination of a stable isotope approach with supplemental methods, such as fatty acid analysis, might be a more beneficial yet more time-consuming and less cost-effective strategy to further assess the consumption of terrestrial/chemosynthetic organic matter by macrofauna.

## ACKNOWLEDGEMENTS

We thank the crew and participants of the 82nd cruise of R/V "Akademik Mstislav Keldysh" for successful sampling in the Laptev Sea.

### Funding

This research was supported by the Russian Science Foundation (grant # 20-74-10011). This research was carried out as part of the Scientific Project of the State Order of the Government of Russian Federation to Lomonosov Moscow State University No. 121032300121-0. Contribution of Olga Konovalova was supported by the Tomsk State University Development Program (Priority-2030). The funders had no role in study design, data collection and analysis, decision to publish, or preparation of the manuscript.

### Grant Disclosures

The following grant information was disclosed by the authors:
Russian Science Foundation: 20-74-10011.
Lomonosov Moscow State University: 121032300121-0.
Tomsk State University Development Program: Priority-2030.

## Competing Interests

The authors declare that they have no competing interests.

## Author Contributions

- Valentin Kokarev conceived and designed the experiments, performed the experiments, analyzed the data, prepared figures and/or tables, authored or reviewed drafts of the article, and approved the final draft.
- Anna K. Zalota performed the experiments, authored or reviewed drafts of the article, and approved the final draft.
- Andrey Zuev performed the experiments, authored or reviewed drafts of the article, and approved the final draft.
- Alexei Tiunov performed the experiments, authored or reviewed drafts of the article, and approved the final draft.
- Petr Kuznetsov performed the experiments, authored or reviewed drafts of the article, and approved the final draft.
- Olga Konovalova performed the experiments, authored or reviewed drafts of the article, and approved the final draft.
- Nadezhda Rimskaya-Korsakova conceived and designed the experiments, performed the experiments, authored or reviewed drafts of the article, and approved the final draft.

## Field Study Permissions

The following information was supplied relating to field study approvals (*i.e.*, approving body and any reference numbers):

Field samplings and experiments were approved by the Ministry of Education and Science of the Russian Federation.

## Data Availability

The raw data are available in the Supplemental Files.

## Supplemental Information

Supplemental information for this article can be found online at http://dx.doi.org/10.7717/peerj.15595#supplemental-information.

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
