# Peer review of "Opportunistic consumption of marine pelagic, terrestrial, and chemosynthetic organic matter by macrofauna on the Arctic shelf: a stable isotope approach"

_PeerJ, doi:10.7717/peerj.15595_

## Round 0.1 · original submission · Major Revisions

Dear Authors,

Please find the reviews of your manuscript. After careful consideration and based on the reviewers’ comments, I have decided that your manuscript requires a major revision before it can be accepted for publication. Further consideration of the manuscript will be contingent upon revision according to the detailed reviewers’ suggestions.

The topic of the current submission is timely, and the presented study is likely to be of interest to the broad scientific community. There are, however, several points that require additional attention. These include reporting the appropriate analytical accuracy and additional statistical description.

Based on the reviewers’ comments, please revise the manuscript considering all suggestions carefully, and either change the manuscript appropriately or provide convincing reasons for declining to do so. Please prepare a detailed, point-by-point list of your replies to the comments of the reviewers. For each reviewer comment, make a clear connection to your response, to facilitate my simultaneous consideration of the reviewers’ comments and your replies to those comments.

Reviewer 1 ·

Basic reporting

The manuscript is well-written, although I suggest a higher focus in the main patterns. Species-specific description of isotope niche is a little exhaustive. Please, reduce the results section by at least 20% or 30%.

- English, literature references, article structure are ok. Hypothesis and results are relevant and contain novelties.

Experimental design

My main concern is the absence of replica of habitats, especially the “background” treatment. Also, the proximity of the background site to seeps is an issue to me. The authors need to argue whether sites are independent of each other, or to acknowledge the partial dependence as a limitation of the study.

-Research is within "aims and scope" of the journal.
-Research questions are clear.
-Investigation was rigorous and with high technical and ethical standard.
-Methods need clarification of some issues raised here and in the pdf file.

Validity of the findings

The results are consistent. I suggest fitting models with interaction between “habitat” and “feeding habitat” instead of Kruskall-Wallis. Also, report the magnitude of effects; in some cases, differences in mean values (e.g., 13C signatures) are very close, but there is statistical support with low magnitude (p~0.02). Please, focus on differences that really have biological meaning. Delta seems the most different habitat in terms of organic matter consumption by macrofauna, and this makes sense when considering the proximity of background and seep habitats.

Additional comments

Please, see my specific comments and suggestions in the pdf file.

Annotated reviews are not available for download in order to protect the identity of reviewers who chose to remain anonymous.

·

Basic reporting

Line 103: I suggest changing “-26 to -28‰” to “-28 to -26‰”.

Line 104: Following the cited literature (Stein and Macdonald, 2004; Tamelander et al., 2009; Bell et al., 2016), “20 to 22‰” may be rewritten as “-22 to -20‰”.

Supplemental materials: Tables 2, 3 and 4 are headed as Table 1. I suggest that the authors fix it.

Experimental design

Lines 164-166: Please provide more details on Frigidoalvania sp. collected in the previous cruise. Was that cruise the AMK-69 (Table 1)? If not, I suggest that you add sampling dates and locations. As you said, these gastropods have a very patchy distribution.

Lines 168 - 169: The method used to remove the shells and desiccate the mollusk foot needs to be clarified. Please, add that information to the sentence.

Lines 177- 178: Please explain why surface sediments were not sampled at stations 6976 and 5625.

Lines 231-233: To apply those cited R tools (mainly SIBER), data must meet homogeneity of variances and normal distribution criteria (especially food sources). Also, food sources must be isotopically distinct from each other. Table 4 shows some very similar values of sediments from different stations. Please provide information about those tests on your data. If possible, please share your R code. Sharing materials encourages reuse and facilitates reproducibility.

Line 267: Please rewrite “diffent habitats” as “different habitats”.

Validity of the findings

Lines 310- 313: As you report: “all the samples from the “Background” for both feeding groups were fully incorporated within the “Seep” community isotopic niche”. Have you considered incorporating “latitude” as a variable? “Background" and “Seep” are extremely near stations (at the same latitude), while “Delta” station is at a lower latitude. Maybe you can run a general linear model (or other stats) to search for a variable that better explains the macrofauna isotopic values (if the location, niche, taxa or station). As reported in Section 3.2, species differentiated from each station, and taxonomy should also be considered here. It may help to distinguish highly overlapped groups (Figure 5 and lines 377 to 379).

Lines 394-408: Unfortunately, without information on sediment stats and trophic enrichment factors for those animals, it is not recommended to affirm that “these species might consume organic matter of different origins depending on availability”. Please consider rewriting that section to fit your findings.

Section 4: The contribution of terrestrial, marine organic matter or chemosynthetic origin to the macrofauna cannot be asserted based on your data. Although higher or lower isotopic values may indicate the contribution of this or that food source, results may be misinterpreted without a relevant sampling that meets models’ criteria. Maybe, you could improve the sampling number using other researchers’ materials and run a mixing model (i.e., MixSIAR). If you agree, please rewrite the whole discussion section without the food source contribution to the macrofauna. If you disagree, please explain why you state that based on your data.
In my opinion, you should focus the manuscript discussion on the macrofauna communities’ isotopic niche distinctiveness at each location, as most of your results and figures show.

Section 5: As pointed out previously, I suggest reviewing the focus of the manuscript from the origin of the food source to the distinct isotopic niche of macrofauna in each location. If you agree, please rewrite the conclusion section.

Additional comments

I commend the authors for their effort to investigate such a remote area. Due to the study area’ challenging hard climate and accessibility conditions, food source materials are difficult to obtain. To infer the proportional source contribution to the consumers, mixing models are highly recommended to be performed on robust data (i.e., MixSIAR - https://doi.org/10.7717/peerj.5096). Without those results, premature conclusions can have massive consequences on the direction of future studies and resources applied to science. Therefore, please consider focusing your findings and conclusions on the macrofauna isotopic niche, as shown in figures and tables.

---

## Round 0.2 · accepted · Accept

Considering, that the authors have corrected or clearly justified the limitations of the study and the relevance of the manuscript, after reading all the comments, I support the publication of this manuscript. Congratulations.